# Knowledge, Attitude, and Practice of Healthcare Providers Towards Preventive Chemotherapy Neglected Tropical Diseases in the Forécariah Health District, Guinea, 2022

**DOI:** 10.3390/tropicalmed9110273

**Published:** 2024-11-11

**Authors:** Fatoumata Diaraye Diallo, Tamba Mina Millimouno, Hawa Manet, Armand Saloum Kamano, Emmanuel Camara, Bienvenu Salim Camara, Alexandre Delamou

**Affiliations:** 1Programme National de Lutte contre les Maladies Tropicales Négligées, Conakry P.O. Box 585, Guinea; assbindiop@yahoo.fr; 2Africa Centre of Center d’Excellence for Prevention and Control of Communicable Diseases (CEA-PCMT), Gamal Abdel Nasser University of Conakry, Conakry P.O. Box 1017, Guinea; hawa@maferinyah.org (H.M.); askamano@maferinyah.org (A.S.K.); dremmano74@gmail.com (E.C.); bscamara@maferinyah.org (B.S.C.); adelamou@cea-pcmt.org (A.D.); 3Department of Public Health, Gamal Abdel Nasser University of Conakry, Conakry P.O. Box 1017, Guinea; 4Centre National de Formation et de Recherche en Santé Rurale de Maferinyah, Forecariah P.O. Box 2649, Guinea; 5Centre Hospitalier Universitaire Donka, Conakry P.O. Box 234, Guinea

**Keywords:** knowledge, attitudes, practices, health providers, neglected tropical diseases, preventive chemotherapy, Guinea

## Abstract

**Background:** Neglected tropical diseases (NTDs) are a diverse group of twenty diseases that occur in tropical and subtropical regions that particularly affect vulnerable and often marginalised populations. Five of these are classified as “preventive chemotherapy” (PC) diseases such as trachoma, onchocerciasis, geo-helminthiasis, lymphatic filariasis, and schistosomiasis. This study aimed to describe the knowledge, attitudes, and practices of healthcare providers in the Forecariah health district with respect to PC-NTDs in Guinea in 2022. **Methods:** A descriptive cross-sectional study was conducted from 7 to 22 November 2022 among healthcare providers in the health district of Forécariah in Guinea. Data on participants’ socio-demographic characteristics and knowledge of and attitudes and practices regarding PC-NTDs were collected using an electronic (KoboToolbox) semi-structured questionnaire and analysed using descriptive statistics. **Results:** Among the 86 healthcare providers who participated in this study, nurses (44.2%) and young adults aged between 25 and 49 years (81.4%) were mostly represented. The majority of respondents declared having already heard about onchocerciasis (70.7%) and lymphatic filariasis (60.0%) but only the minority declared having already heard about geo-helminthiasis (30.7%), schistosomiasis (21.3%), and trachoma (9.3%). Only a few respondents knew how to prevent PC-NTDs (onchocerciasis 26.7%, lymphatic filariasis 26.7%, geo-helminthiasis 29.3%, and schistosomiasis 17.3%). Many healthcare providers reported they would refer cases of onchocerciasis (50.6%), lymphatic filariasis (58.7%), and schistosomiasis (46.7%) to a management centre. **Conclusions:** This study highlights the varying levels of knowledge, attitudes, and practices among healthcare providers in dealing with PC-NTDs, suggesting areas for improvement in training and resource allocation.

## 1. Introduction

Neglected tropical diseases (NTDs) are a diverse group of twenty diseases that occur in tropical and subtropical regions and particularly affect vulnerable and often marginalised populations [1]. According to the World Health Organization (WHO), NTDs affect more than one billion people worldwide, with more than 500,000 deaths per year [1]. Furthermore, Africa carries the highest burden of NTDs, with 17 priority diseases, five of which are classified as “preventive chemotherapy” (PC) diseases, including trachoma, onchocerciasis, geo-helminthiasis, lymphatic filariasis, and schistosomiasis [2]. Some of these PC-NTDs are parasitic, such as onchocerciasis, geo-helminthiasis, lymphatic filariasis and schistosomiasis. Thus, the WHO has put in place an accelerated plan of programmatic action that includes the scaling up of cross-cutting approaches and modification of the operational model to facilitate country ownership from 2014 to 2020. The goal of this plan is to reduce the magnitude of NTDs and their disease-related and socio-economic burdens in affected populations [1].

Despite the remarkable progress made in the fight against NTDs over the past decades, NTDs still persist in sub-Saharan Africa and the targets that were set for 2020 have not been met [1]. Factors linked to the non-achievement of the targets included low levels of training of health providers, limited access to effective drugs, stigma, socio-cultural beliefs, and prejudices surrounding the diseases.

In Guinea, eight of the twenty NTDs identified by the WHO are endemic in the country’s 33 health districts [3]. For example, national surveys conducted between 2011 and 2017 reported a considerable prevalence of NTDs (10% for schistosomiasis, 11% for lymphatic filariasis, 15% for onchocerciasis, and 66.7% for geo-helminthiasis) [3]. To reduce the transmission and morbidity of these NTDs, several approaches have been initiated, such as the mass treatment of the five PC-NTDs from 2014 and treatment with ivermectin under community guidelines in 2022 [3]. In addition, community-based services provided by community health workers (CHWs), community outreach workers, and primary healthcare services are essential supports for the prevention and management strategies of NTDs.

Data on PC-NTDs exist in Guinea and are limited to prevalence surveys, clinical aspects, and management or documentation of the implementation of interventions [4,5,6]. There is one study that reported on the knowledge, attitudes, and practices (KAP) of the general population on lymphatic filariasis, which was conducted in an urban, non-endemic area (Conakry) [6]. There is therefore a need to explore the knowledge, attitudes, and practices of primary health care providers across a wider range of PC-NTDs in rural endemic areas. Thus, this study aimed to describe the knowledge, attitudes, and practices of health care providers in the Forécariah health district with respect to PC-NTDs in Guinea in 2022. Specifically, the study sought to (1) describe the socio-demographic characteristics of the health providers participating in the study, (2) describe the knowledge, attitudes, and practices of the health providers regarding PC-NTDs, and (3) explore the existence of a link between the socio-demographic characteristics of the providers and their (levels of) KAP regarding NTDs. The identification of the KAP levels of primary healthcare providers in relation to PC-NTDs will help to support and guide current strategies to achieve the WHO’s recommended goal of the elimination of these diseases by 2030.

## 2. Materials and Methods

### 2.1. Study Design

This was an analytical cross-sectional study conducted from 7 to 22 November 2022 in the health district of Forécariah in Guinea.

### 2.2. Setting

#### 2.2.1. General Setting

Guinea is a west African country with an estimated population of 12,559,623 in 2020, the majority of whom (70%) live in rural areas [7]. The national health care system is organised into three levels, including the primary level consisting of 407 health centres and 1640 health posts, the secondary level consisting of seven regional hospitals, 26 prefectural hospitals, and nine communal medical centres, and the tertiary level consisting of three national hospitals [7].

#### 2.2.2. Specific Setting

The Forécariah health district is in the Lower Guinea region and borders Sierra Leone. It had an estimated population of 298,264 in 2020 and includes a prefectural hospital, an improved health centre, 11 health centres (including one urban), and 52 health posts [7]. The 11 health centres and one health post were the study sites. The choice of these sites was based on the fact that they were at the primary level, constituting the gateway to the health care system, and were the structures for the implementation of PC strategies. In terms of endemicity in the country, there are five PC-NTDs among the eight endemic NTDs, four of which (onchocerciasis, geo-helminthiasis, lymphatic filariasis, and schistosomiasis) are prevalent in the health district of Forécariah, placing the district as one of the most endemic in the country [3].

### 2.3. Study Population

The study population consisted of healthcare providers from health centres and health posts involved in the prevention, referral, and management of patients suffering from NTDs or those who had participated in mass distribution campaigns of ivermectin, praziquantel, or albendazole. All healthcare providers at health centres and health posts who were present on the day of the survey and who agreed to be interviewed were included in the study. These included medical doctors, nurses, and community health workers. Nurses and community health workers undergo a three-year training program in health schools, receiving theoretical and practical education in various health topics, including infectious diseases such as NTDs. The difference between them resides in the basic education level required to access the training and the training content and work packages. One of the eligibility criteria to access the nursing training is to successfully complete senior secondary school, while those who access the HCWs training are required to successfully complete junior secondary school (Grade 10). Nurses receive more extensive formal training, while community health workers typically focus on community outreach and prevention at the primary care level. Both nurses and CHWs are expected to support mass drug administration campaigns, health promotion, and basic disease management.

#### 2.3.1. Sampling

The sampling was done at two levels (health centre level and provider level). For the health centre level, we used non-random sampling. All 11 health centres and one health post in the Forécariah health district were included in the study. We conducted an exhaustive sampling of all healthcare providers who met our inclusion criteria (as stated earlier).

#### 2.3.2. Variables

Data variables included: socio-demographic characteristics of the participants (age, gender, type of health facility, level of education, occupation, and length of service); general knowledge of health providers on NTDs (knowledge on PC-NTDs, types of NTDs on which providers had knowledge, endemic NTDs in Guinea, endemic NTDs in Forécariah, PC-NTDs, receipt of training on PC-NTDs, type of NTDs on which providers were trained, number of healthcare providers trained on NTDs per year from 2015 to 2022, specific knowledge of healthcare providers on onchocerciasis, lymphatic filariasis, schistosomiasis, and geo-helminthiasis in terms of causal agent, mode of transmission, clinical signs, prevention, and notification/referral of cases in 2021); providers’ attitudes towards a PC-NTD case (feeling confidence in managing PC-NTDs, belief in specialised care for patients, and belief in the necessity of moral support for patients and their parents); and providers’ practices towards an PC-NTD case (referred cases, sensitised cases, cases confirmed by a consultation/examination, examination performed, cases treated/cared for, use of an NTD case reporting form, nature of the service receiving the notification, feedback to services on notified cases, involvement in the PC campaign, and number of times of involvement in PC campaigns).

### 2.4. Data Collection

Data were collected using a semi-structured questionnaire that was pre-tested and administered individually to the eligible providers in the study. The questionnaire was structured according to the study objectives and integrated into the electronic data collection system (KoboToolbox). Questions were developed by the research team based on their knowledge of the NTDs and the study objectives. A team of two research assistants who were general practitioners collected the data under the supervision of the study’s principal investigator.

### 2.5. Data Management and Analysis

The data extracted from the KoboToolbox were processed with Excel (version 2019) and then imported into Stata 16.1 (Stata Corporation, College Station, TX, USA) for analysis. The results were presented in the form of descriptive statistics (numbers, proportions, and medians with interquartile ranges).

### 2.6. Ethical Considerations

This study was approved by the Comité National d’Ethique pour la Recherche en Santé en Guinée (N°147/CNERS/22 dated 18 October 2022). In the field, we obtained approval from the Forécariah health district authorities and written consent from all participants. No personal/identifying information was collected during data collection from the participants and confidentiality of information was maintained throughout the study.

## 3. Results

### 3.1. Sociodemographic Characteristics

A total of 86 healthcare providers participated in the study, with a median age of 31 years (range 25–64 years). Young adults aged between 25 to 49 years were the most represented at 81.4% and women represented 54.6%. More than three-quarters of the respondents (97.7%) were from health centres. More than half of the providers had a professional level of education, at 63.9%, and 44.2% were nurses. Seniority between one and five years was the most common form of experience, at 69.8% (Table 1).

### 3.2. Healthcare Providers’ Knowledge of PC-NTDs

Healthcare providers’ knowledge of PC-NTDs is shown in Table 2. More than three quarters of the providers had heard of PC-NTDs (87.2%). Of these, onchocerciasis, lymphatic filariasis, and geo-helminthiasis were mentioned by 70.7%, 60.0%, and 30.7%, respectively.

Specifically, most providers said they had received training on PC-NTDs (58.7%). Among those who had received training, 84.1% had received training on onchocerciasis and 34.1% had received training on schistosomiasis. Though the training of providers on NTDs started in 2015, they were mostly trained in 2021 (40.0%). Most providers did not know the disease’s causal agents (81.3% for onchocerciasis, 78.7% for lymphatic filariasis, 80% for schistosomiasis, and from 74.7% to 93.3% for geo-helminthiasis). In terms of prevention of onchocerciasis, few providers (26.7%) mentioned the yearly administration of ivermectin. As a preventive method, most providers cited the use of mosquito nets (26.7%) for lymphatic filariasis, avoiding contaminated water (17.3%) for schistosomiasis, and washing your hands frequently (29.3%) for geo-helminthiasis.

### 3.3. Knowledge of Onchocerciasis

Few respondents (18.7%) knew the causative agent of onchocerciasis. As for the mode of transmission, human-to-human transmission (29.3%) followed by the bite of an infected simulium black fly (18.7%) were the most cited. Regarding clinical signs, decreased vision, irreversible blindness, and itching were the most listed at 36.7%, 28.0%, and 24.0%, respectively. Prevention of the disease by distributing ivermectin every year was mentioned by only 20 respondents, or 26.7%.

### 3.4. Knowledge of Lymphatic Filariasis

Concerning the KAP of lymphatic filariasis, 21.3% of the respondents knew the causal agent of the disease. Among the causal agents cited, mosquitoes and filaria were cited by 68.7% and 12.5% of respondents, respectively. When asked about the mode of transmission, mosquito bites and human-to-human transmission were reported.

Fever and elephantiasis of the lower and upper limbs, breasts, and scrotum were mentioned as clinical signs. Regarding prevention, the use of mosquito nets was cited with a frequency of 26.7%, followed by albendazole (5.3%) and the use of repellent creams (4.0%).

### 3.5. Knowledge of Schistosomiasis

Of the 75 respondents, 26.7% knew the causative agent of the disease. When asked about the mode of transmission, contaminated water and faeces were mentioned by 21.3% and 5.3%, respectively. Regarding clinical signs, the presence of blood in the urine was the most cited sign (24.0%), followed by the presence of blood in stools (20.0%).

As for prevention, avoiding contaminated water and mass treatment were the most listed measures, with frequencies of 17.3% and 16.0%, respectively.

### 3.6. Knowledge of Geo-Helminthiasis

Ascariasis (25.3%) and hookworm (16.0%) were the most frequently encountered causative agents, while whipworms and pinworms were mentioned by 4.0%.

Regarding the mode of transmission, walking barefoot (24.0%) and eating soiled food (20.0%) were also cited. Frequent diarrhoea (25.3%), abdominal pain (16.0%), and abdominal bloating (12.0%) were the main clinical signs reported.

When asked about the mode of prevention, regular hand washing (29.3%), avoiding walking barefoot (12.0%), and use of latrines (9.3%) were the main modes of prevention cited by our respondents.

### 3.7. Attitudes of Healthcare Providers Towards Preventive Chemotherapy Parasitic NTDs

More than half of the healthcare providers interviewed reported that they do not feel confident in managing onchocerciasis (57.3%), lymphatic filariasis (64.0%), and schistosomiasis (52.0%). A considerable proportion of healthcare providers (33.3–52.0%) believe that parasitic NTDs require specialized care. Some providers also mentioned that moral support is needed for patients and their parents (2.6–9.3%) (Table 3).

### 3.8. Practice of Healthcare Providers in Dealing with Preventive Chemotherapy Parasitic NTDs in Forécariah Health District, Guinea, 2022 (n = 75)

Most healthcare providers reported referring cases of parasitic NTDs: onchocerciasis (50.6%), lymphatic filariasis (58.7%), schistosomiasis (46.7%), and geo-helminthiasis (38.7%). A small proportion of health providers used a reporting or notification form for parasitic NTD cases (16.0%). Of the 12 providers who had reporting forms, the majority (16.6%) sent the notification to the district health office. A small proportion (16.0%) of the providers received feedback from the services receiving the case notification (84.0%). Less than half (48.0%) of the participants were involved in preventive chemotherapy campaigns and the majority (69.4%) of these participants reported having been involved once (Table 4).

## 4. Discussion

Our study is one of the few to report on the knowledge, attitudes, and practices of healthcare providers toward PC-NTDs in Guinea. The results highlight several key areas that require attention to improve the overall prevention and management of PC-NTDs. These are awareness of PC-NTDs, training and knowledge gaps, clinical manifestation and prevention, and attitudes and practices. Below, we discuss some of these areas.

### 4.1. Awareness of PC-NTDs

More than half of the healthcare providers were aware of onchocerciasis and lymphatic filariasis. This result might be attributable to the widespread endemicity of both diseases in 24 districts of Guinea [1]. In areas where these diseases are widespread and public health campaigns are actively addressing them, healthcare providers are more likely to be informed. The Ethiopian data corroborates this, as the high prevalence rates (6.9–85.3%) of onchocerciasis likely drive awareness efforts [8]. However, our results contrast with Nigerian data [6], where 82.1% of participants were unaware of lymphatic filariasis despite its endemicity. The lower awareness of lymphatic filariasis among Nigerian participants, despite its endemicity, suggests potential gaps in public health communication, training, or healthcare system responsiveness to neglected tropical diseases (NTDs). This contrast could be due to differences in how national health programs prioritize these diseases or how effectively they disseminate information to healthcare workers. These variations imply that effective public health strategies depend heavily on localised efforts. In areas where awareness is lower, despite high endemicity, like Nigeria, there may be a need for increased educational campaigns, better healthcare provider training, and more robust disease surveillance systems to improve the understanding and management of NTDs.

Most healthcare providers were not familiar with schistosomiasis, geo-helminthiasis, or trachoma. This lack of knowledge in our study could be due to the delayed integration of lymphatic filariasis treatment in mass drug administration campaigns and insufficient training for healthcare providers. This suggests that the healthcare providers, especially those in health centres, may not be sufficiently informed or equipped to address a broader range of NTDs, which could hinder effective prevention and treatment efforts. This finding deviates from studies by Laurentine et al. [9], Yusof et al. [10], and Nath et al. [11], which reported a high awareness of geo-helminthiasis. Evidence from these studies implies that in other regions, healthcare systems have successfully integrated geo-helminthiasis and other PC-NTDs into their public health frameworks and training programs. This highlights the need for comprehensive and uniform training on all five PC-NTDs to ensure that healthcare providers can address the full range of endemic diseases effectively.

### 4.2. Training and Knowledge Gaps

Most participants had received training on PC-NTDs, with over three-quarters trained on onchocerciasis and fewer than half on schistosomiasis. This is likely because onchocerciasis is endemic in many districts in Guinea [12], and annual mass drug distribution campaigns involve healthcare providers from these regions.

However, knowledge about transmission modes was generally poor across all PC-NTDs. For instance, more than three-quarters of those aware of onchocerciasis did not know that the black fly is its vector, similar to findings in Cameroon [9]. Additionally, many participants were unaware that mosquitoes transmit lymphatic filariasis, a trend observed in other regions as well. The knowledge of schistosomiasis and geo-helminthiasis transmission was similarly low, echoing findings from Sacolo et al. [13]. These knowledge gaps could be attributable to limited training opportunities and varying exposure to NTD-related resources among healthcare providers, and these gaps can hinder effective disease prevention and control efforts. If healthcare providers are unaware of how these diseases are transmitted, they may be less effective in educating patients and communities about preventive measures. The educational background of most of our study participants, particularly the nurses and community health workers, should enable them to have at least a basic understanding of NTDs, including their transmission, prevention, and treatment. However, our findings on knowledge gaps may indicate that while NTDs are covered in their curricula, there could be issues with the depth of understanding, retention of information, or the effectiveness of the training programs.

### 4.3. Clinical Manifestations and Prevention

Regarding clinical manifestations, over half of the participants did not recognize eye disease as a symptom of onchocerciasis, unlike the majority in Laurentine et al.’s study [9] who identified eye disease due to the common name “river blindness”. This suggests a potential gap in understanding the full clinical spectrum of onchocerciasis among healthcare providers in our setting. The lack of recognition of eye disease, which is a hallmark symptom of onchocerciasis, could indicate a need for more targeted education and awareness campaigns about the condition. Healthcare providers may not be adequately trained to recognize the disease’s diverse manifestations, particularly its visual impacts.

Conversely, most healthcare providers identified elephantiasis as a symptom of lymphatic filariasis, differing from Amaechi et al. [14], who reported swellings as the most common symptom in Nigeria. The differences in symptom recognition between our study and others highlight the importance of localising health education to ensure healthcare providers are equipped to recognise both early and severe disease symptoms. Improving early detection skills would enable more timely interventions, reducing the burden of these diseases in endemic regions [1].

Awareness of clinical signs for schistosomiasis and geo-helminthiasis was relatively high, with most participants identifying blood in urine and stools for schistosomiasis, and abdominal pain and frequent diarrhoea for geo-helminthiasis. These findings could indicate that targeted campaigns emphasising the most common and visible clinical signs have had an impact on healthcare provider knowledge. They, however, contrast with Gwebu et al. [13], who noted significant misconceptions about symptoms, suggesting that public health interventions and education efforts are not uniform across regions. It emphasises the need for context-specific approaches to education and training to address local knowledge gaps. Misconceptions may persist in settings where public health messaging has not effectively reached or engaged healthcare providers, underlining the importance of tailored interventions [15].

Additionally, more than half of the study participants knew how to prevent schistosomiasis and geo-helminthiasis. These findings align with those of Nath et al. [15] in Bangladesh, where 64.4% of school-aged children reported preventing soil-transmitted helminthiasis (STH) by washing hands after defecation and 75.6% knew that taking medication was necessary to control STH. This consistency across studies suggests that basic prevention messages for NTDs, such as hygiene practices and medication, are being effectively communicated and understood in various settings. This highlights the importance of reinforcing simple, actionable health behaviours as part of public health campaigns, which can have a wide-reaching impact across different regions and populations [16].

Surprisingly, most respondents did not know that annual ivermectin distribution prevents onchocerciasis, despite government campaigns involving healthcare providers. This contrasts with a Nigerian study [17], where a high percentage (88.4%) of respondents knew that ivermectin is a preventive drug for onchocerciasis. Similarly, few participants were aware of lymphatic filariasis prevention methods, unlike in Nigeria [18], where a significant proportion knew about prevention and treatment. This is further underscored by the contrast in knowledge levels observed in other regions, such as Nigeria and Bangladesh, where awareness and understanding of preventive measures are higher. In our setting, despite annual government campaigns, the lack of awareness about the symptoms and prevention methods for diseases like onchocerciasis and lymphatic filariasis suggests current training and educational initiatives are insufficient. Possible issues could include insufficient outreach, ineffective messaging, or a lack of engagement with local communities. This highlights the need to evaluate and enhance public health campaigns, ensuring they effectively communicate the importance of preventive measures and reach their intended audience.

To improve disease management and control, there is an urgent need for enhanced, targeted training programs that address these knowledge gaps, especially regarding disease vectors, clinical signs, and effective prevention strategies. Enhanced supervision and integration of lymphatic filariasis treatment into mass drug administration programs are also critical.

#### Limitations

The main limitations of this study are as follows: (1) the use of a non-random sampling method at the health centre level may limit the generalizability of the findings to other regions or healthcare settings (sampling bias), (2) the reliance on self-reported questionnaires may introduce social desirability or recall bias, potentially affecting the accuracy of the data on healthcare providers’ knowledge, attitudes, and practices (self-reported data), (3) and conducting the study solely in the Forécariah health district restricts the applicability of the results to other regions with different healthcare infrastructures or levels of NTD endemicity (limited geographical coverage). These limitations should be considered when interpreting the results of the study and when planning future research in this area. However, our results are transferrable to other health districts of the country with similar socio-cultural characteristics and disease endemicity.

## 5. Conclusions

Our study reveals significant disparities in the knowledge, attitudes, and practices of healthcare providers in the Forécariah health district regarding neglected tropical diseases requiring preventive chemotherapy. Although the majority of providers have some familiarity with diseases such as onchocerciasis and lymphatic filariasis, a significant proportion of them demonstrate gaps in knowledge about diseases like schistosomiasis and geo-helminthiasis, as well as in the modes of transmission, clinical signs, and preventive measures. These deficiencies are also reflected in their limited confidence in managing cases of NTD-PC and in their practices of referring such cases to specialized centres. These results underscore the urgent need to strengthen continuous training for healthcare providers, particularly on the full range of NTD-PCs, in order to improve the prevention and treatment of these diseases. Based on the above, health authorities should (1) reinforce continuous training for healthcare providers, with a particular focus on the modes of transmission, clinical signs, and preventive methods for NTD-PCs, and (2) encourage additional research to understand the observed disparities in the knowledge, attitudes, and practices of healthcare providers.

## Figures and Tables

**Table 1 tropicalmed-09-00273-t001:** Sociodemographic characteristics of the participants in the KAP study on “preventive chemotherapy” neglected tropical diseases (PC-NTDs), Forécariah, Guinea, 2022 (N = 86).

Variables	Number	Percentage
Age (years)		
15–24	10	11.6
25–49	70	81.4
50–64	6	7.0
Median (IQR)	31 (26–39)	
Gender		
Women	47	54.6
Men	39	45.3
Types of health facility		
Health centre	84	97.7
Health post	2	2.3
Level of education		
Secondary	20	23.3
Professional	55	63.9
Academic	11	12.8
Profession (n = 85)		
Nurses	38	44.2
Community health workers	34	39.5
Medical doctors	8	9.3
Biologists	3	3.5
Midwivese	2	2.3
Seniority (years)		
<1	10	11.6
1–5	60	69.8
>5	16	18.6

IQR: Interquartile range.

**Table 2 tropicalmed-09-00273-t002:** Healthcare providers’ knowledge of “preventive chemotherapy” neglected tropical diseases (PC-NTDs), in Forécariah health district, Guinea, 2022 (n = 75).

Variables	Number	Percentage
Have knowledge of PC-NTDs		
Yes	75	87.2
No	11	12.8
Type of PC-NTDs on which participants had knowledge (n = 75)		
Onchocerciasis	53	70.7
Lymphatic filariasis	45	60.0
Schistosomiasis	16	21.3
Geo-helminthiasis	23	30.7
Trachoma	8	9.3
Training on PC-NTDs (n = 75)		
Yes	44	58.7
No	31	41.3
PC-NTDs on which participants were trained (n = 44 for each modality below)		
Onchocerciasis	37	84.1
Lymphatic filariasis	35	79.5
Schistosomiasis	15	34.1
Onchocerciasis (n = 75)		
Mode of transmission		
Infected simulium black fly bite	14	18.7
Running river	3	4.0
Human-to-human transmission	22	29.3
Clinical Signs		
Decreased vision	29	38.7
Irreversible blindness	21	28.0
Itching	18	24.0
Nodule	4	5.3
Dry skin	2	2.7
Skin rash	12	16.0
Leopard skin	4	5.3
Fever	13	17.3
Watery eyes	1	1.3
Light gene	1	1.3
Lymphatic filariasis		
Mode of Transmission		
Mosquito bite	18	24.0
Sick person to a healthy person	4	5.3
Insect bite	1	1.3
Clinical signs		
Fever	10	13.3
Elephantiasis of the upper limbs	12	16.0
Elephantiasis of the lower limbs	30	40.0
Elephantiasis of the scrotum	6	8.0
Elephantiasis of the breast	13	17.3
Schistosomiasis		
Mode of transmission		
By faeces	4	5.3
Through contaminated water	16	21.3
By the urine of a patient	2	2.7
Clinical Signs		
Blood in urine	18	24.0
Blood in the stool	15	20.0
Frequent diarrhoea	4	5.3
Abdominal pain	12	16.0
Abdominal bloating	12	16.0
Geo-helminthiasis		
Mode of transmission		
Human excrement	5	6.7
Walking barefoot	18	24.0
Soiled water	11	14.7
Eating soiled food	15	20.0
Clinical Signs		
Frequent diarrhoea	19	25.3
Abdominal pain	12	16.0
Physical Asthenia	1	1.3
Anal itching	3	4.0
Abdominal bloating	9	12.0

**Table 3 tropicalmed-09-00273-t003:** Attitudes of healthcare providers towards parasitic neglected tropical diseases, in Forécariah health district, Guinea, 2022 (n = 75).

Variables	Number	Percentage
Dealing with onchocerciasis		
Do not feel confident in managing onchocerciasis	43	57.3
Believe that onchocerciasis requires specialised care	29	38.6
Believe that moral support is needed for patients and their parents	3	3.9
Dealing with lymphatic filariasis		
Do not feel confident in managing filariasis	48	64.0
Believe that filariasis requires specialised care	25	33.3
Believe that moral support is needed for parents	2	2.6
Facing schistosomiasis		
Do not feel confident in managing schistosomiasis	39	52.0
Believe that schistosomiasis requires specialised care	22	42.7
Believe that moral support is needed for patients and their parents	4	5
Dealing with geo-helminthiasis		
Do not feel confident in managing geo-helminthiasis	29	38.7
Believe that geo-helminthiasis requires specialised care	39	52.0
Believe that moral support is needed for patients and their parents	7	9.3

**Table 4 tropicalmed-09-00273-t004:** Provider practices for parasitic neglected tropical diseases, in Forécariah Health District, Guinea, 2022.

Variables	Number	Percentage
Onchocerciasis-related practices:		
Referred cases	38	50.6
Sensitised cases	7	9.3
Seek to confirm the diagnosis	1	1.3
Do not know	25	33.3
Lymphatic filariasis-related practices:		
Referred cases	44	58.7
Sensitised cases	2	2.7
Seek to confirm	1	1.3
Do not know	28	37.3
Schistosomiasis-related practice		
Referred cases	35	46.7
Cases treated/cared for	10	13.3
Stool examination performed	2	2.7
Seek to confirm	1	1.3
Do not know	27	36.0
Geo-helminthiasis-related practices:		
Referred cases	26	34.7
Cases treated/cared for	19	25.3
Stool examination performed	2	2.7
Confirmed cases	1	1.3
Do not know	27	36.0
Use case reporting/notification template		
Yes	12	16.0
No	63	84.0
Nature of the service receiving the notification (n = 12)		
Neglected tropical diseases research centre	2	16.7
Health district office	8	66.7
Hospital teaching university	1	8.3
Preventive chemotherapy	1	8.3
Feedback to services on notified cases		
Yes	12	16.0
No	63	84.0
Involvement in preventive chemotherapy campaigns		
Yes	36	48.0
No	39	52.0
Number of times involved in preventive chemotherapy campaigns (n = 36)		
Once	25	69.4
Twice	11	30.6

## Data Availability

The database used in this study is available upon request from the corresponding author.

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
