# Peer review of "Knowledge, Attitude, and Practice of Healthcare Providers Towards Preventive Chemotherapy Neglected Tropical Diseases in the Forécariah Health District, Guinea, 2022"

_tropicalmed, 2024, doi:10.3390/tropicalmed9110273_

Round 1
Reviewer 1 Report
Comments and Suggestions for Authors
This is a small study on the knowledge, attitudes and practices of healthcare workers in a district in Guinea. While the approach is not very innovative and the sample is rather small, the majority of primary healthcare workers in the district were included, and the study gives important insight into the situation on the ground in this district. The manuscript is suitable for the special issue on NTDs in West Africa, after revision.
The characteristics of the study population is given too much emphasis in the abstract – this could be shortened, and more KAP data be presented in the abstract.
Chapter 3.7 Attitudes (and also respective paragraph in discussions section): the authors describe practices, rather than attitudes here. Is there any more detailed information on attitudes of healthcare workers in the given context? It would be interesting having some information of the attitudes of the healthcare workers regarding the diseases in question.
The discussion focuses on comparisons of the data from this study with other studies, without a thorough discussion on the findings and their implications, as compared to findings from other authors and regions. The authors may revise the discussion section likewise.
The limitations section could include a brief statement on external validity – it is said that the study was only done in one health district, but can the data be extrapolated to the other districts in the country (same socio-cultural and endemicity characteristics?).
Author Response
REVISEUR 1
Commentaires et suggestions pour les auteurs
Il s’agit d’une petite étude sur les connaissances, les attitudes et les pratiques des agents de santé dans un district de Guinée. Bien que l’approche ne soit pas très innovante et que l’échantillon soit plutôt petit, la majorité des agents de santé primaires du district ont été inclus, et l’étude donne un aperçu important de la situation sur le terrain dans ce district. Le manuscrit est adapté au numéro spécial sur les MTN en Afrique de l’Ouest, après révision.
Rep: Thanks for your positive comments
Les caractéristiques de la population étudiée reçoivent trop d’importance dans le résumé – celui-ci pourrait être raccourci et davantage de données CAP pourraient être présentées dans le résumé.
Rep: Thanks for this comment. Your suggestion has been taken into account
Chapitre 3.7 Attitudes (et également paragraphe correspondant dans la section discussions) : les auteurs décrivent ici des pratiques plutôt que des attitudes. Existe-t-il des informations plus détaillées sur les attitudes des professionnels de santé dans le contexte donné ? Il serait intéressant d'avoir des informations sur les attitudes des professionnels de santé à l'égard des maladies en question.
Rep: Thanks for these comments. We agree with your suggestions, and we have revised this chapter accordingly.
La discussion porte sur les comparaisons des données de cette étude avec celles d'autres études, sans discussion approfondie des résultats et de leurs implications, par rapport aux résultats d'autres auteurs et régions. Les auteurs peuvent également réviser la section de discussion.
Rep: Thanks for these comments. We have now strengthened the discussion section following your remarks.
La section sur les limites pourrait inclure une brève déclaration sur la validité externe – il est dit que l’étude n’a été réalisée que dans un seul district sanitaire, mais les données peuvent-elles être extrapolées aux autres districts du pays (mêmes caractéristiques socioculturelles et d’endémicité ?).
Rep: Thanks for this suggestion. We added a sentence about the transferability of the findings.
Reviewer 2 Report
Comments and Suggestions for Authors
The authors intended to evaluate knowledge, attitude and practice of health care providers on neglected tropical diseases in several sites of Guinea. 86 health care providers were concerned and were responding to a questionnaire. It was performed in November 2022.
The previous formation of health care providers is very important. It is reduced to secondary, professional and academic. I guess that nurse, midwife, biologist and medicine doctors (rather than doctors) are the professional ones. Community health workers are difficult to define: what kind of teaching did they receive? This should be clarified since the overall knowledge on tropical neglected diseases is apparently poor and thus, I wonder how the health care centre can really help the population. These points should absolutely be clarified.
Are the health care providers giving general advice or providing dugs? I did not clearly see what their task is. It should be explained. It could be interesting to see if nurses and health care providers had a different level of knowledge on these diseases.
The questions posed to health care providers are numerous. How were they chosen? This is also quite important since questions are originating from knowledge and beliefs of the interviewers and may bias the results. Why the interviewers were dermatologist or general practionner?
Are Schistosomas, only S. haematobium or also S. mansoni, since the questions may be different.
The knowledge on geo-helminths seems very low. How can it be explained since worms such Ascaris or tapeworms are easily seen? Enterobius are also causing nearly specific symptoms.
What is the interest of presenting median age and interquartile range? This would make sense if the sample was randomly taken.
The tables are replete with figures, and I think they are not needed since they are clearly presented again in text.
The latin names of parasites should be in italics (see table 2 for example).
The acknowledgements are lengthy and might be reduced to the essential.
The references must be checked carefully. The first one is without author or origin, The second has also a problem. The third had three dots and I do not know why. And mistakes are on the other references as well. No 12 is also wrong with internet connection.
Author Response
REVISEUR 2
Commentaires et suggestions pour les auteurs
Les auteurs se sont attachés à évaluer les connaissances, l'attitude et la pratique des prestataires de soins sur les maladies tropicales négligées dans plusieurs sites de Guinée. 86 prestataires de soins ont été concernés et ont répondu à un questionnaire. L'enquête a été réalisée en novembre 2022.
Rep: Thanks for this kind introductory note
La formation préalable des prestataires de soins est très importante. Elle se réduit à une formation secondaire, professionnelle et académique. Je suppose que les professionnels sont les infirmiers, les sages-femmes, les biologistes et les médecins (plutôt que les médecins). Les agents de santé communautaires sont difficiles à définir : quel type d'enseignement ont-ils reçu ? Cela devrait être clarifié car les connaissances générales sur les maladies tropicales négligées sont apparemment faibles et je me demande donc comment le centre de soins peut réellement aider la population. Ces points devraient absolument être clarifiés.
Rep: Thanks for your comments. As shown in Table 1, participants included medical doctors (n=8), but the majority were nurses (n=38) and community health workers (n=34) with secondary/professional levels of education. In Guinea, Nurses and community health workers undergo a three-year training program in schools of health, receiving both theoretical and practical education in various health topics, including infectious diseases such as NTDs. They are expected to support mass drug administration campaigns, health promotion, and basic disease management.
This background in education suggests that most of our study participants, particularly the nurses and CHWs, should have at least a basic understanding of NTDs, including their transmission, prevention, and treatment. However, our findings on knowledge gaps may indicate that while NTDs are covered in their curricula, there could be issues with the depth of understanding, retention of information, or the effectiveness of the training programs.
We added some clarifications to the study population and the discussion sections (page 11, lines 291-296).
Les soignants donnent-ils des conseils généraux ou des conseils ? Je n'ai pas bien vu quelle était leur tâche. Il faudrait l'expliquer. Il pourrait être intéressant de voir si les infirmières et les soignants avaient un niveau de connaissances différent sur ces maladies.
Rep: Thanks for your comments. All healthcare providers including medical doctors, nurses and community health workers were involved in providing general advice and drugs. This has been added to the study population section.
Les questions posées aux professionnels de santé sont nombreuses. Comment ont-elles été choisies ? C'est également un point très important car les questions proviennent des connaissances et des croyances des enquêteurs et peuvent biaiser les résultats. Pourquoi les enquêteurs étaient-ils dermatologues ou généralistes ?
Rep: Thanks for your comments. Questions were developed by the research team based on their knowledge of the NTDs and the study objectives. Data collectors were general practitioners. This clarification has been added to the paper (data collection section)
Les Schistosomas sont-ils uniquement S. haematobium ou également S. mansoni, car les questions peuvent être différentes.
Rep: Thanks for your comments. Sorry, we did not explore that information.
Les connaissances sur les géo helminthes semblent très limitées. Comment expliquer cela alors que les vers comme les Ascaris ou les ténias sont facilement visibles ? Les Enterobius sont également responsables de symptômes presque spécifiques.
Rep: Thanks for your comments.
Quel est l'intérêt de présenter l'âge médian et l'écart interquartile ? Cela aurait du sens si l'échantillon était prélevé de manière aléatoire.
Rep: Thanks for your comments. Presenting the median age and interquartile range (IQR) in a study, even when the sample is not randomly selected, has several benefits, particularly for data description and interpretation.
Les tableaux sont remplis de chiffres, et je pense qu'ils ne sont pas nécessaires puisqu'ils sont clairement présentés à nouveau dans le texte.
Rep: Thanks for your comments. We have removed some unnecessary figures from the Tables.
Les noms latins des parasites doivent être en italique (voir tableau 2 par exemple).
Rep: Thanks for this comment. We have made corrections.
Les remerciements sont longs et pourraient être réduits à l’essentiel.
Rep: Thanks for this comment. We have made corrections.
Il faut vérifier soigneusement les références. La première est sans auteur ni origine, la deuxième a aussi un problème. La troisième avait trois points et je ne sais pas pourquoi. Et il y a aussi des erreurs sur les autres références. Le numéro 12 a aussi un problème de connexion Internet.
Rep: Thanks for this comment. We have revised the references
Round 2
Reviewer 2 Report
Comments and Suggestions for Authors
The paper intends to evaluate knowledge, attitude and practices regarding neglected tropical diseases. The paper is improved from previous review but the main problems remain.
NTD are not only parasitic since leprosy, rabies, trachoma, Buruli ulcer, dengue and chikungunya among others are included. The authors should add” parasitic” to NTD. They mention trachoma in some parts of the paper, which is not parasitic.
The healthcare providers are medicine doctors, midwives, biologists, nurses and community health workers. What is the difference between the two last ones? It is strange not to compare knowledge of these different workers since it would be expected that their knowledge is different.
This study shows the lack of knowledge of healthcare providers relative to parasitic neglected tropical diseases. The authors do not give real explanation of this lack of knowledge. The attitudes and practices are not studied in detail, but the main problem remains lack of knowledge.
The discussion is lengthy and could be shortened. The table are extensive and, in some cases, a brief text description could be enough.
Author Response
Comments and Suggestions for Authors
The paper intends to evaluate knowledge, attitude and practices regarding neglected tropical diseases. The paper is improved from previous review but the main problems remain.
Response: Thank you for recognizing the improvements made in the paper. We acknowledge that some key issues remain. We have now carefully reviewed the main points raised to address these effectively.
NTD are not only parasitic since leprosy, rabies, trachoma, Buruli ulcer, dengue and chikungunya among others are included. The authors should add” parasitic” to NTD. They mention trachoma in some parts of the paper, which is not parasitic.
Response: We appreciate the suggestion regarding the terminology of “parasitic” neglected tropical diseases (NTDs). While most NTDs are parasitic, trachoma is not. We have clarified this distinction by specifying “parasitic NTDs” where relevant in the manuscript to avoid confusion.
The healthcare providers are medical doctors, midwives, biologists, nurses and community health workers. What is the difference between the two last ones? It is strange not to compare the knowledge of these different workers since it would be expected that their knowledge is different.
Response: Thank you for highlighting the need to distinguish between nurses and community health workers. The difference between them resides in the basic education level to access the training and the training content and work packages. One of the eligibility criteria to access the nursing training is to successfully complete the senior secondary school while those who access the HCWs training are required to successfully complete the junior secondary school (Grade 10). Nurses receive more extensive formal training, while community health workers typically focus on community outreach and prevention at the primary care level.
We agree that comparing the knowledge of the different health workers could reveal valuable insights. However, this was out of the scope of this work.
This study shows the lack of knowledge of healthcare providers relative to parasitic neglected tropical diseases. The authors do not give a real explanation of this lack of knowledge. The attitudes and practices are not studied in detail, but the main problem remains the lack of knowledge.
Response: We appreciate this observation regarding the lack of knowledge. We have added a more thorough explanation in the discussion, exploring possible reasons for the observed knowledge gaps, such as limited training opportunities and varying exposure to NTD-related resources among healthcare providers.
The discussion is lengthy and could be shortened. The table are extensive and, in some cases, a brief text description could be enough.
Response: Thank you for your suggestion to streamline the discussion and tables. We have shortened the discussion by removing the section on attitudes and practices, and some detailed data from table 2 to a narrative format to enhance readability.
Round 3
Reviewer 2 Report
Comments and Suggestions for Authors
The paper is ok. There are some minor mistakes:
1) the plural of miwife is midwives (in table)
2) references:
No 10 has authors' names in upper case. Write them in lower case.
No 15 Santé OM is not convenient and differs from other references where it is mentioned as Organisation mondial de la santé (OMS)
3) Conclusion title appears twice.